# Monitoring *E. coli* Cell Integrity by ATR-FTIR Spectroscopy and Chemometrics: Opportunities and Caveats

Jens Kastenhofer 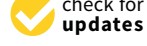, Julian Libiseller-Egger, Vignesh Rajamanickam and Oliver Spadiut *

Research Group Integrated Bioprocess Development, Research Division Biochemical Engineering, Institute of Chemical, Environmental and Bioscience Engineering, TU Wien, Gumpendorfer Strasse 1a, 1060 Vienna, Austria; jens.kastenhofer@tuwien.ac.at (J.K.); julian.libiseller-egger@hotmail.com (J.L.-E.); vignesh.rajamanickam@boehringer-ingelheim.com (V.R.)
* Correspondence: oliver.spadiut@tuwien.ac.at; Tel.: +43-1-58801-166473

**Abstract:** During recombinant protein production with *E. coli*, the integrity of the inner and outer membrane changes, which leads to product leakage (loss of outer membrane integrity) or lysis (loss of inner membrane integrity). Motivated by current Quality by Design guidelines, there is a need for monitoring tools to determine leakiness and lysis in real-time. In this work, we assessed a novel approach to monitoring *E. coli* cell integrity by attenuated total reflection Fourier-transform infrared (ATR-FTIR) spectroscopy. Various preprocessing strategies were tested in combination with regression (partial least squares, random forest) or classification models (partial least squares discriminant analysis, linear discriminant analysis, random forest, artificial neural network). Models were validated using standard procedures, and well-performing methods were additionally scrutinized by removing putatively important features and assessing the decrease in performance. Whereas the prediction of target compound concentration via regression was unsuccessful, possibly due to a lack of samples and low sensitivity, random forest classifiers achieved prediction accuracies of over 90% within the datasets tested in this study. However, strong correlations with untargeted spectral regions were revealed by feature selection, thereby demonstrating the need to rigorously validate chemometric models for bioprocesses, including the evaluation of feature importance.

**Keywords:** bioprocess monitoring; ATR-FTIR spectroscopy; quality by design; process analytical technology; chemometrics; machine learning



## 1. Introduction

Soluble expression of recombinant protein in *Escherichia coli* is often achieved by translocation of the product to the periplasm, the space between the inner membrane (IM) and outer membrane (OM). For successful bioprocessing aligned with Quality by Design and process analytical technology principles, the integrity of both membranes needs to be monitored in real-time to allow timely decision-making and control [1]. Protein leakage through the OM to the extracellular space has a strong impact on the downstream process, either as a way for selective product release and simplified primary recovery or as unwanted loss of product. In any case, the ability to distinguish changes in OM integrity (leakiness) from a loss of IM integrity (lysis) is paramount. Lysis does not only lead to reduced productivity but also to the release of impurities, such as host cell protein (HCP), DNA, and lipids, along with the product, thereby affecting the purification process.

A variety of technologies that have the potential for monitoring IM and OM integrity exist and were recently reviewed [1]. However, actual use cases in *E. coli* bioprocesses are still scarce. Apart from directly measuring membrane properties (dielectric spectroscopy) or using dyes for assessing their permeability (flow cytometry), one approach to monitoring IM and OM integrity entails measuring cytoplasmic compounds (protein, nucleic acids) or membrane components in the culture supernatant as a proxy for lysis as well as periplasmic protein (native or recombinant) as an indicator for leakiness.

Mid-infrared (MIR; 2.5–25 μm or 4000–400 cm$^{-1}$) spectroscopy is potentially applicable to this purpose. Many biomolecules display specific spectral features in the MIR range, particularly in the fingerprint region (900–1800 cm$^{-1}$) [2,3]. Thus, on-line MIR spectroscopy provides valuable information for bioprocess monitoring [4], although up-stream applications have mostly been focused on the quantification of small organic metabolites or inorganic medium components so far [5–9]. However, the fact that the secondary structure and amino acid composition affect the protein spectrum has been exploited to distinguish the product from HCP in downstream applications [10–12]. Real-time sampling in the bioreactor may be achieved by in-line attenuated total reflectance (ATR) probes. These sensors can be inserted into the vessel for direct contact with the sample. Thus, no additional liquid handling is necessary. Furthermore, they can be cleaned and sterilized in place, reducing contamination issues. In an ATR element, the IR light is subjected to one or multiple total reflections at the boundary layer of a crystal with high optical density, such as a diamond, and the analyte solution [13,14]. At the reflection interface, an evanescent field emerges and penetrates the sample, where absorbance occurs. Commonly, the penetration depth is in the range of ~1–2 μm, and the absorbance decays exponentially with increasing depth. Therefore, the contribution of biomass to the absorbance spectra is minimal when using an ATR probe in a stirred culture [15,16]. This may facilitate monitoring of leakiness and lysis of *E. coli* cultures by measuring changes in the composition of the culture supernatant. On one hand, this may be achieved by predicting concentrations of specific marker analytes by multivariate regression, such as extracellular periplasmic product for leakiness or extracellular HCP or DNA for lysis. Alternatively, leakiness and lysis may be defined as cell phases that serve as target variables in multivariate classification tasks.

Partial least squares regression (PLSR) is the most commonly employed method for bioprocess monitoring [17], whereas non-linear models (such as support vector machines with non-linear kernels, artificial neural networks (ANN), or random forest regression (RFR)) have been used due to their ability to better capture the non-linearity of cell cultures [18–21]. Supervised classification tasks are less common for real-time bioprocess monitoring, but they have been applied for the detection of metabolic phases during yeast fermentation [19,22]. Common classification models for spectral data comprise partial least squares discriminant analysis (PLSDA), linear discriminant analysis (LDA), support vector machines, random forest classifiers (RFCs), or ANN [22–24]. The optimization of chemometric models usually consists of multiple iterations of data preprocessing, selection of features, and model complexity, followed by appropriate validation procedures. General components and workflows have been reviewed [17,24–29]. Model validation should ultimately ensure robust predictions based on causal correlations [17]. Rigorous validation is particularly important for bioprocess data due to high collinearity between multiple process variables as well as process parameters and time, which may be reflected in the spectral features [7]. Thus, feature selection may not only be used as a means for model enhancement (i.e., removing unimportant features to reduce overfitting) but also for model validation by removing putatively important variables and assessing whether prediction performance decreases accordingly.

This work assessed the applicability of in-line attenuated total reflection Fourier-transform infrared (ATR-FTIR) spectroscopy for real-time monitoring of the OM and IM integrity of *E. coli* during recombinant protein production. After exploring the spectra of various defined samples off-line (e.g., of pure medium components), we performed in-line measurements of six controlled lab-scale fed-batch bioprocesses and analyzed the spectra with various computational and chemometric methods. These included (i) different preprocessing strategies, (ii) multivariate regression by PLSR and RFR, as well as (iii) classification by PLSDA, LDA, RFC, and ANN. Standard validation metrics were employed, and well-performing models were additionally scrutinized by feature selection. Whereas regression did not yield results viable for monitoring, classification by RFC had high apparent prediction accuracies of over 90%. However, feature selection revealed that predictions were strongly based on spurious correlations. This demonstrated that such

feature selection steps should be implemented in validation procedures to question the performance of chemometric models for bioprocessing.

## 2. Materials and Methods

### 2.1. ATR-FTIR Off-Line Spectra

A ReactIR 45m spectrometer and a 1.5 m silver halide optical fiber ATR probe with a diamond tip (Mettler Toledo, Columbus, OH) were used to record MIR spectra between 650 and 3000 cm$^{-1}$ at a resolution of 4 cm$^{-1}$. A matching software to record data was employed (iC IR 4.2, Mettler Toledo). The detector was cooled with liquid nitrogen, and the spectrometer was constantly purged with dry air in order to minimize water vapor artifacts. Off-line spectra of medium components, cell suspensions (whole and lysed), culture supernatant, and bovine serum albumin (BSA) were recorded by immersing the ATR probe into a stirred beaker with the respective solutions at room temperature. Additionally, the spectrum of biomass was determined by allowing cells to settle on the surface of the ATR probe in an upright position. The cell dry weight concentration of cell suspensions was 29 g/L. Lysis was performed at 1000 bar for 3 passages in a PandaPLUS 2000 high-pressure homogenizer (GEA, Düsseldorf, Germany). A spectrum of water and/or buffer was taken directly before measuring the relevant solution in order to account for changes in baseline between replicates, as small alterations in the orientation of the fiber conduit can impact the measurement.

### 2.2. Bioreactor Cultivations and In-Line ATR-FTIR Measurements

Six bioreactor experiments were performed in the course of this study, named LC1, LC2, LC3, LC4, LC5, and LC6. The used strain was *E. coli* X-press, a proprietary strain developed by enGenes Biotech [30,31], harboring a plasmid containing the sequence for recombinant staphylococcal protein A (SpA) [32]. The strain was cultivated as previously described [32]. Briefly, a 500 mL pre-culture was grown overnight in a semi-defined medium, and a 100 mL aliquot was used to inoculate 900 mL defined minimal medium with 20 g/L glucose in a DASGIP system (Eppendorf, Hamburg, Germany) with 2 L working volume kept at 37 °C. Dissolved oxygen was kept above 30% by adjusting addition of pure oxygen. The stirrer speed was constant at 1200 rpm. After the initial glucose was depleted, minimal feed medium with 400 g/L glucose was fed to the culture exponentially to reach biomass concentrations of ~30 g/L. Then, expression of SpA was induced by adding IPTG and L-arabinose to final concentrations of 0.5 and 100 mM, respectively. L-arabinose is needed for the decoupling of growth and recombinant protein production in *E. coli* X-press, whereas IPTG induces the plasmid-based expression [30]. The temperature was reduced to 30 °C during the induction phase, and a constant feed rate was set to achieve a theoretical specific growth rate of 0.05 for LC4 and LC6 or 0.1 h$^{-1}$ for all other runs, assuming a constant yield coefficient $Y_{X/S}$ of 0.4 g g$^{-1}$.

For the in-line ATR-FTIR measurements, background spectra were acquired in the air once the probe was in its final orientation. A spectrum was recorded every five minutes with 256 scans each.

### 2.3. Reference Off-Line Measurements

Extracellular SpA and activity of extracellular alkaline phosphatase (AP) were quantified as indicators of leakiness, whereas extracellular DNA was measured as a proxy for lysis. The analysis methods have been described elsewhere [32]. Samples were taken (i) after inoculation, (ii) at the end of the batch phase, (iii) directly before and after induction, and (iv) subsequently in intervals between one and three hours until the end of the process. A total of 67 reference samples were collected (18 pre-induction, 49 post-induction samples) (Figure S1).

### 2.4. Programming

Data analysis was performed in Python 3.8. Tools for basic signal processing and statistical analyses were available via SciPy v.1.5.2 [33] and NumPy v.1.19.2 [34]. Dimension reduction, classification, and regression were performed with scikit-learn v.0.23.2 [35] and the Keras API upon TensorFlow v.2.3.0 [36,37].

### 2.5. Preprocessing and Validation Split

The fingerprint region (1700–850 cm$^{-1}$) was selected, and spectra were further pre-processed using various strategies. The first and second derivatives of the raw spectra (denoted d1 and d2, respectively) were taken via the Savitzky–Golay smoothing process with third-order polynomials and a window length of 11. To accommodate for between-process variability in the collected backgrounds, baseline-corrected spectra were generated by smoothing the spectra with a Savitzky–Golay filter (without derivative, d0) and subsequently subtracting the mean of the first five spectra from all spectra of the corresponding process. Additionally, the first and second derivatives of the baseline-corrected spectra were taken. Finally, the baseline-correction strategy with and without derivatives was applied exclusively to spectra recorded after induction. In summary, nine sets of processed spectra were candidates for subsequent modeling: (i) raw [d0] (as reference); (ii) raw [d1]; (iii) raw [d2]; (iv) baseline-corrected [d0]; (v) baseline-corrected [d1]; (vi) baseline-corrected [d2]; (vii) baseline-corrected (post-induction) [d0]; (viii) baseline-corrected (post-induction) [d1]; and (ix) baseline-corrected (post-induction) [d2].

Principal component analysis (PCA) was used to explore the effect of the preprocessing methods on within-run and between-run variability and select a preprocessing strategy suitable for further modeling. Prior to PCA, the spectra were mean-centered and scaled to unit variance.

For subsequent regression and classification tasks, the data were split six-fold using five of the six runs for training and holding out the remaining run for external validation. This ensured independence of the validation set by accounting for between-run variability.

### 2.6. Multivariate Regression

Regression was performed by PLSR (PLS1, NIPALS-algorithm) and RFR for the prediction of extracellular SpA, AP, and DNA. Whereas SpA may accumulate in amounts that can be detected by the ATR-FTIR spectrometer (several grams per liter), it was acknowledged that AP and DNA could merely serve as proxies for released periplasmic and cytoplasmic/cellular compounds, respectively, since their mass concentration in the medium is likely below the detection limit. Only the spectra recorded at times when reference samples were taken were used for regression. For PLSR, the spectra and reference data were centered, scaled, and the tested number of components ranged from one (lowest complexity) to 20 (highest complexity). For RFR, the number of trees in the forest was set to 200, and the pruning parameter was varied between 0.0 (highest complexity) and 0.3 (lowest complexity) in steps of 0.03. The normalized root mean square error (NRMSE) was used as the validation metric (Equation (1)).

$$NRMSE = \frac{\sqrt{\sum_{i=1}^{n} \frac{(\hat{y}_i - y_i)^2}{n}}}{y_{max} - y_{min}} \tag{1}$$

$n$, number of samples; $\hat{y}$, predicted value; $y$, measured value; $y_{max}$, maximal measured value; $y_{min}$, minimal measured value.

The root mean square error is a common metric for regression and min-max normalization allows comparison between differently scaled variables [7,17]. The number of samples was equal to all available samples from all runs. Hence, the NRMSE was averaged over all six external validation splits by concatenating the respective vectors of measured and predicted values.



### 2.7. Classification

Supervised classification was completed by first assigning the labels normal, leaky, or lysis to the spectra based on the reference data (Figure S1). Cells were assumed leaky when the measured concentration of extracellular SpA exceeded 0.5 g/L and assumed to lyse when extracellular DNA concentration exceeded 15 mg/L. Due to a large data gap in run LC5, the threshold for lysis was estimated by linear interpolation. The employed classification methods were PLSDA, LDA, RFC, and ANN. Between 1 and 50 components were tested for PLSDA (PLS2, NIPALS-algorithm) and 11 equidistant shrinkage values between 0 and 1 for LDA. The RFC contained 250 trees, and 11 equidistant pruning parameters between 0 and 0.3 were tested. The ANN (a multilayer perceptron with backpropagation) consisted of an input layer and a normalization layer (for centering and scaling the data), a varying hidden layer structure with relu activation functions, and an output layer activated by the softmax function. Hyperparameter tuning entailed (i) testing different hidden layer structures with one, two, or three hidden layers and 10, 50, 100, 200, or 400 nodes each (totaling 15 structures); (ii) varying the number of iterations over the training data (epochs: 25, 50, 75, 100, 200, or 400); and (iii) testing the parameter $\alpha$ for weight penalization (l2 regularization) with values $10^{-10}$, $10^{-8}$, $10^{-6}$, $10^{-4}$, and $10^{-2}$. The adam optimization algorithm and the log loss function were employed for training the ANN, and the batch size for each epoch was set to 32.

Two validation metrics were used for all classification models: firstly, the accuracy score, which expresses the overall fraction of correct classifications; secondly, the $F_1$ score, which is a balanced performance metric combining precision and recall for every class [38]. Accuracy was averaged over all runs and weighted by the number of samples in each run. The $F_1$ score for each class was averaged over all runs and weighted by the class frequency in each run (Table S1).

For RFC models, feature importance was assessed by the mean decrease (Gini-)impurity (MDI) after choosing the pruning parameter resulting in the highest classification performance. The MDI describes how much a variable contributes to a reduction in node impurity (in this case, the Gini impurity) in all trees of the random forest. For a detailed explanation of the MDI, the reader is referred to reference [39]. The MDI for all features (i.e., wavenumbers) was compared to known spectral features of pure components (e.g., amide bands, carbohydrate fingerprint), if available. To investigate the importance of certain spectral regions for model accuracy, subsets of features were then selected manually after preprocessing (Table 1). In general, features with the highest MDI and a putative link to known analytes (e.g., protein, lysed cells, carbohydrates) were removed, whereas others were retained. The spectral window was then narrowed down further in several steps to remove features that displayed smaller MDI values. After each feature selection step, LDA, RFC, and ANN models were trained and evaluated again.

**Table 1.** Feature selection strategies.

| Subset Name | Retained Features (Wavenumbers (cm$^{-1}$)) | Comment |
|---|---|---|
| ref | 850–1700 | reference; whole fingerprint region |
| sel-1 | 1505–1700 | retain only amide I and II region |
| sel-2 | 1080–1700 | remove part of carbohydrate, phosphate, and lysed cells fingerprint |
| sel-3 | 850–1505 | remove amide I and II region |
| sel-4 | 1080–1505 | sel-2 and sel-3 combined |
| sel-5 | 1190–1450 | remove amide I and II region, full carbohydrate and phosphate fingerprint |
| sel-6 | 1190–1360 | sel-5 and remove putative amino acid band |

**Table 1.** *Cont.*

| Subset Name | Retained Features (Wavenumbers (cm$^{-1}$)) | Comment |
|---|---|---|
| sel-7 | 1255–1450 | based on MDI only |
| sel-8 | 1255–1360 | retain only putative amide III region |
| sel-9 | 1360–1450 | retain only putative amino acid band |

## 3. Results

### 3.1. Off-Line Spectra

The concentration of medium components (phosphate buffer, sugars), cells, and protein changes during fermentation. To determine the spectral features corresponding to these compounds and thus aid subsequent analysis of in-line recorded spectra, ATR-FTIR spectra were first collected off-line. Spectra of the major medium components (water, L-arabinose, glucose, and phosphate) are depicted in Figure 1. The absorption of water dominates the fingerprint region with strong peaks between 1500 and 1700 cm$^{-1}$ as well as below 1000 cm$^{-1}$. The bands of the medium components overlap significantly between 950 and 1200 cm$^{-1}$. Strong absorption of these compounds may impede modeling attempts, especially since phosphate and L-arabinose are both diluted at a constant rate after induction, introducing spectral variability that is proportional to process time.

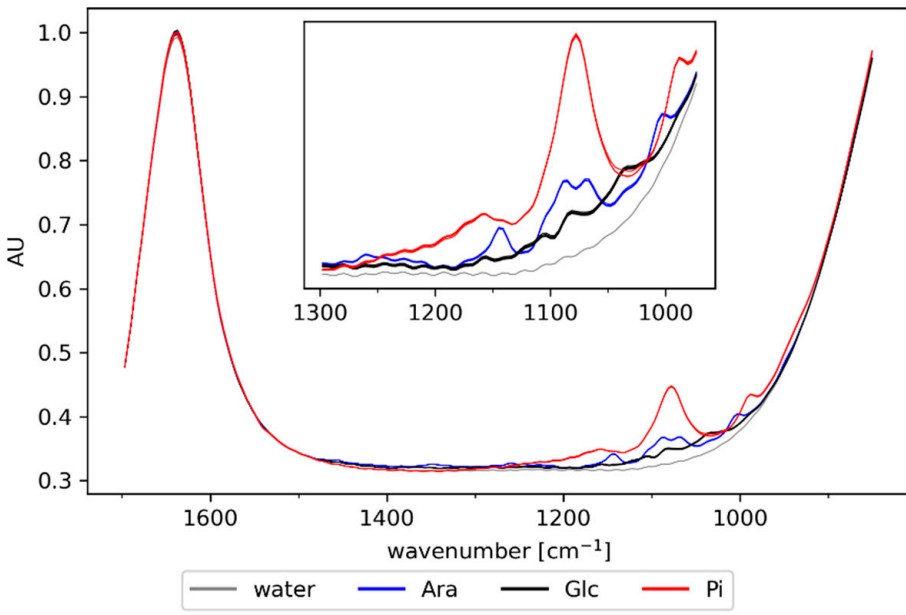

**Figure 1.** Off-line spectra of major medium components: water, L-arabinose (Ara; 100 mM), glucose (Glc, 55 mM), and phosphate (P$_i$, 130 mM). Air was subtracted as background.

The spectral contribution of cells was also assessed in off-line experiments. After placing a drop of cell suspension on the sensor in an upright position, the cells settled quickly onto the ATR crystal. This revealed distinct absorbance at the amide I and II bands (1655 and 1550 cm$^{-1}$, respectively) as well as at 1077 cm$^{-1}$, which may be attributed to cellular carbohydrates [16] (Figure 2A). The increase in absorption across the whole spectral range is attributed to scatter effects. In contrast to the spectra of settling cells, the impact of biomass was much weaker when the spectra were recorded with the ATR probe immersed in a stirred cell suspension (Figure 2B). This was as expected since the penetration depth of the evanescent field, in which absorbance occurs, is only ~0.05–2 μm and thus too small for a large number of suspended cells to be close enough to the ATR element and significantly

affect the signal. The results indicate that the in-line ATR probe facilitates quantification of analyte changes in the extracellular space to monitor leakiness and lysis.

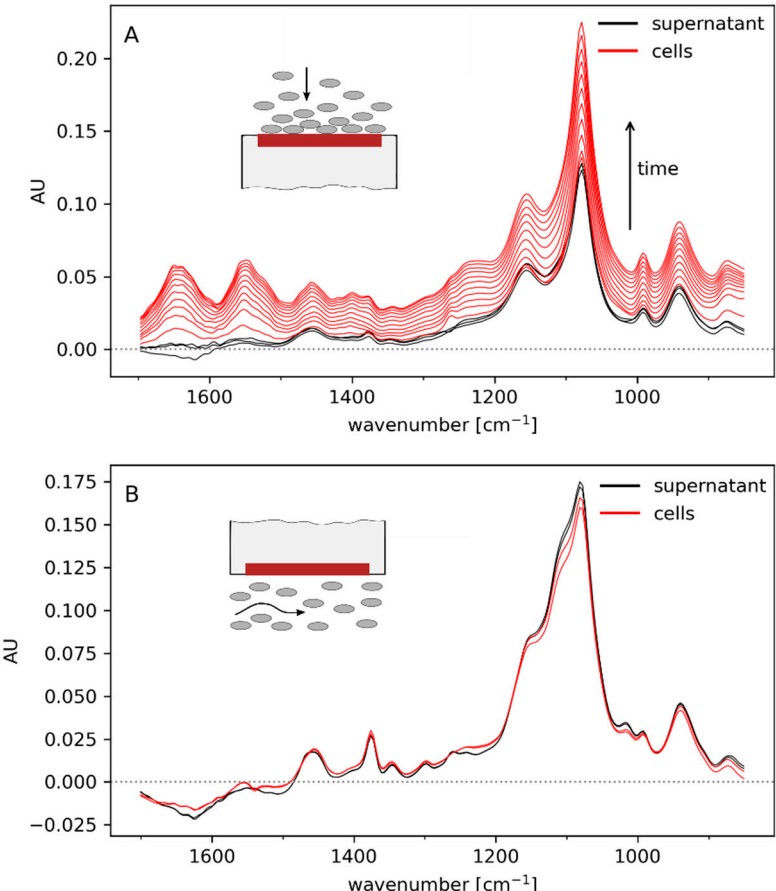

**Figure 2.** Off-line spectra of cells settling on the sensor (**A**) and in a stirred cell suspension (**B**). Cells were suspended in exhausted culture medium, and cell dry weight was 29 g/L. Water was subtracted as background.

Spectra of BSA solutions (ranging between 0 and 20 g/L) were recorded to assess the method's sensitivity to dissolved protein as well as important spectral features (Figure 3A). The most distinct peaks can be attributed to the amide bands I and II (1655 and 1550 cm$^{-1}$), which are known as the most prominent regions of protein spectra [40,41]. Moreover, the weaker amide III band can be identified at 1310 cm$^{-1}$. The remaining bands might be caused by IR-active amino acid side chains (e.g., carboxyl groups in Asp and Glu absorb around 1400 cm$^{-1}$ [40]). However, it should be considered that IR spectra can vary from protein to protein. This is not only due to the different side chains but also because the secondary structure has an impact on the amide bands [40]. This explains the different positions of the amide I bands of BSA (1655 cm$^{-1}$; Figure 3A) and the settling biomass (1644 cm$^{-1}$; Figure 2A). Therefore, distinguishing periplasmic from cytoplasmic protein in the supernatant might be possible, thereby detecting leakiness or lysis. Still, the overlap of the amide bands with the much stronger water peak may reduce method sensitivity. The absolute absorption values of the amide peaks were rather low compared to peaks at lower wavenumbers (e.g., phosphate or carbohydrate peaks). Furthermore, noise introduced by the absorption of water was particularly visible in the amide I band, which displayed almost the same intensity for samples containing 2 and 5 g/L BSA, respectively. In contrast, the amide II band increased proportionally to BSA concentration.

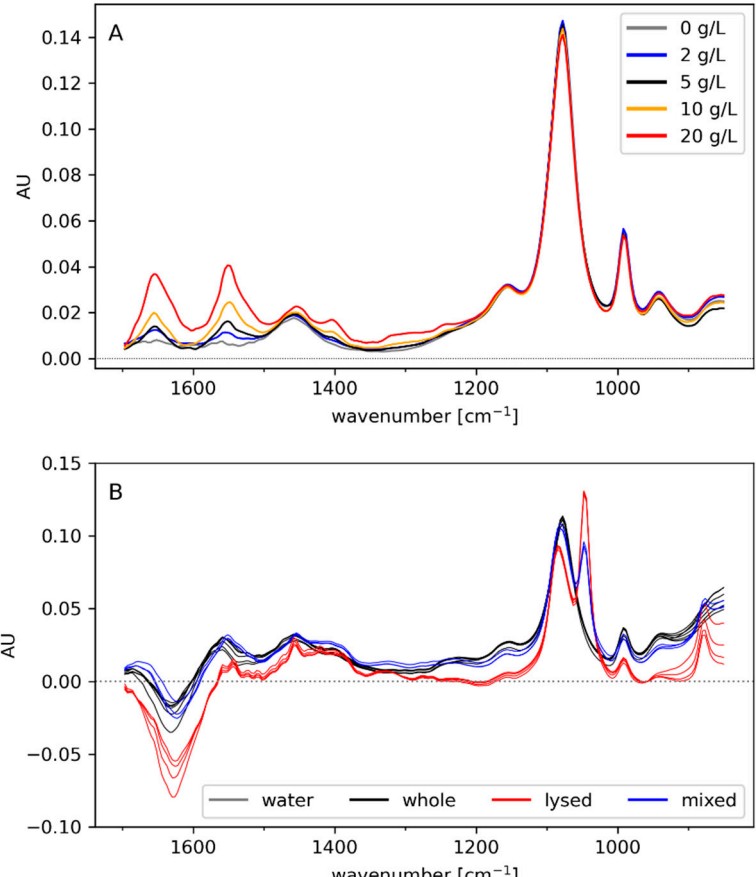

**Figure 3.** (**A**) Off-line spectra of varying concentrations of BSA in phosphate buffer (130 mM). (**B**) Off-line spectra of suspensions of whole cells, lysed cells, or a 1:1 mix of whole and lysed cells. Water was subtracted as background.

In order to assess whether cell lysis can be detected by IR spectroscopy, measurements of whole cells (29 g/L CDW), lysed cells (29 g/L CDW), and of a 1:1 mixture were performed (Figure 3B). Unexpectedly, the amide I band at 1640 cm$^{-1}$ showed lower absorption in the lysed cell sample compared to whole cells. The reason for this remains unclear, though interference of a high amount of solutes with the water peak is possible. The lysed cells displayed distinct absorption at 1048 cm$^{-1}$, likely attributed to C-O/C-O-C vibrations from released membrane components [42,43]. Despite the characteristic fingerprint of artificial cell lysate of relatively high concentration (~15–30 g/L CDW), detection of minor levels of cell lysis during fermentation may be hampered by the overlap of spectral features with the medium components (Figure 1).

### 3.2. Preprocessing

The main goal of spectral preprocessing was to remove the between-run variability (e.g., stemming from measurement artifacts or slight deviations in the measurement setup) and to preserve the within-run variability associated with specific process events, such as leakiness or lysis, for detection by regression or classification models. Unsupervised clustering by PCA was initially used to illustrate the effects of the preprocessing strategies (Figure 4).

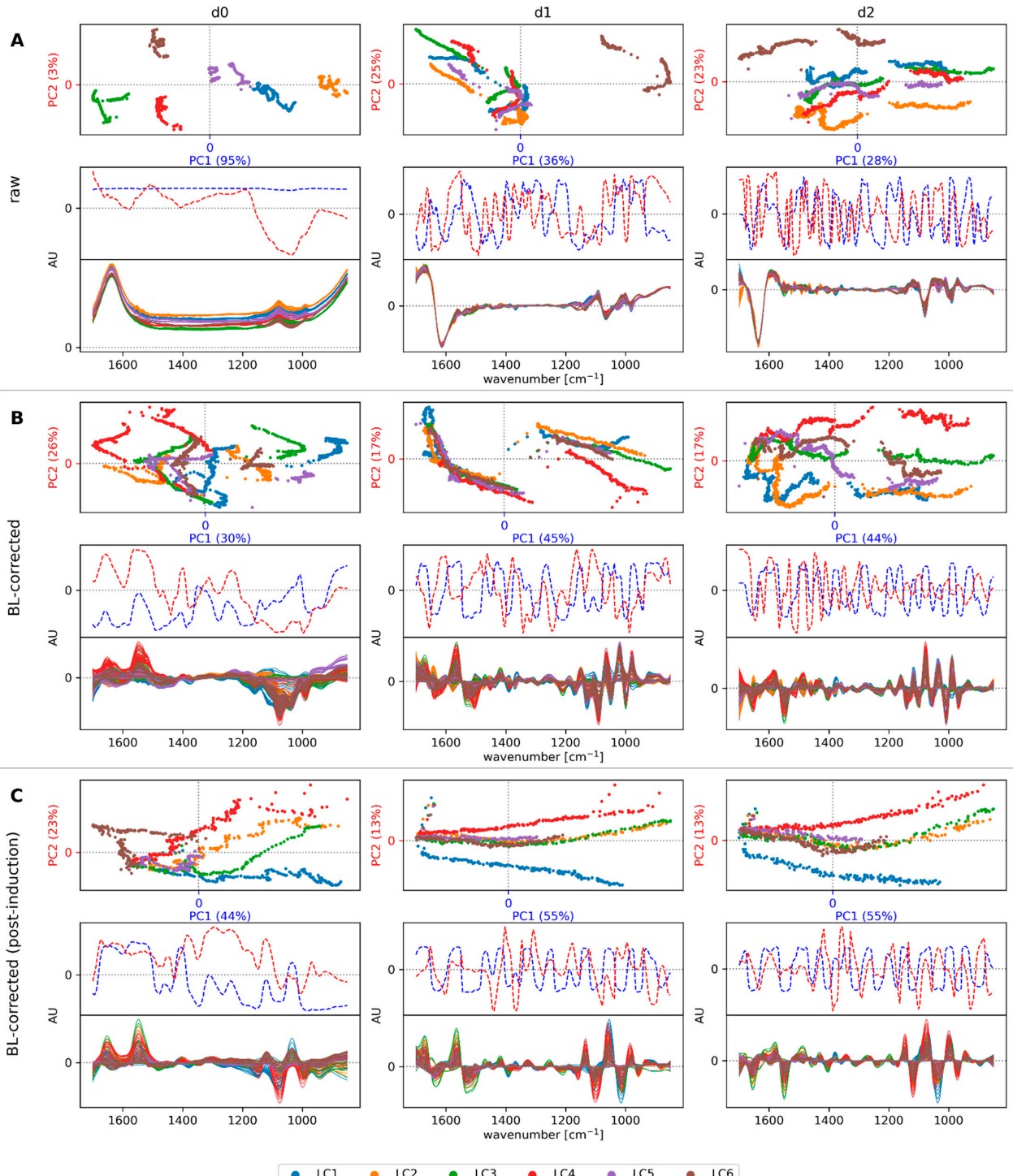

**Figure 4.** Principal component analysis (PCA) of raw spectra (**A**), baseline (BL)-corrected spectra (**B**), and BL-corrected post-induction spectra (**C**). The left column contains undifferentiated spectra (d0), the center column first-derivative spectra (d1), and the right column second-derivative spectra (d2). The top panel of each sub-figure (**A**–**C**) shows the PCA scores with the percentage of variance explained by the respective principal component given in parentheses. The center panel shows the PCA loadings, and the bottom panel the IR spectra. Colors represent the different runs.

Large between-run variability was visible in the raw spectra of the different cultivations, as their baselines were offset vertically. This resulted in an almost uniform loading vector of the first principal component and a clear separation between the processes in the PCA score plot. These baseline offsets were eliminated by first- and second-order

derivatives. However, the different cultivations were still separated, and particularly, run LC6 was distinct from the other runs due to more pronounced differences in the spectral background. Only by applying the baseline-correction strategy (Figure 4B) could we significantly enhance the within-run variability in relation to the between-run variances, especially in combination with the first derivative. Furthermore, each process displayed two main clusters, one before and one after induction. This is due to the addition of 100 mM L-arabinose and a temperature shift upon induction. This large shift in the spectra may particularly hamper classification tasks since most of the spectra labeled normal were recorded before induction, and thus the spectral shift would introduce bias. Therefore, we additionally assessed the baseline-correction strategy on the post-induction spectra alone. As shown in Figure 4C, this resulted in a further decrease of the between-run variability, and within-run variability was largely explained by the first principal component. However, clear separation of the spectra attributed to the different cell states normal, leaky, and lysis did not occur, and the transition between these states was rather smooth (Figure 5). This indicates that most variability was caused by other process-related changes of the spectral fingerprint, which may complicate subsequent modeling efforts.

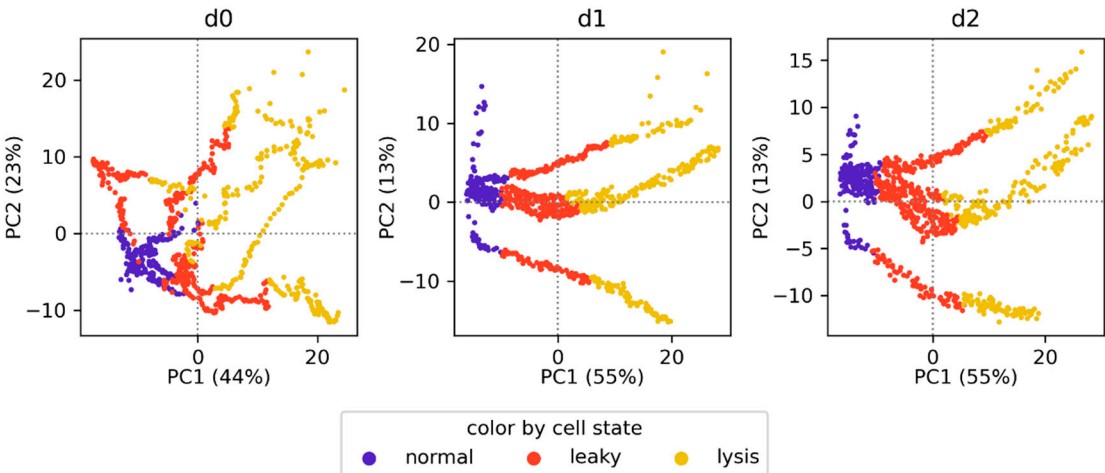

**Figure 5.** Baseline-corrected post-induction spectra color-coded by cell state. Parentheses show the percentage of variance explained by the respective principal component. The columns show zero-order (d0), first-order (d1), and second-order (d2) derivatives.

### 3.3. Regression

Both regression models, PLSR and RFR, were trained with baseline-corrected spectra, either from the whole process or post-induction, for the prediction of extracellular AP, SpA, and DNA. The external validation strategy using independent runs for testing ensured that the models did not learn the potential between-run variability of the test set. The NRMSE values of models with an optimized number of components (PLSR) or pruning parameter (RFR) are summarized in Table 2. Either strategy resulted in high errors between 0.10 and 0.35, rendering regression unsuitable for monitoring purposes. This was likely due to the scarcity of the reference data (67 and 49 samples for the whole process and post-induction, respectively) and the necessary reduction of the spectral dataset for training the models. Furthermore, the sensitivity to target analytes was likely diminished by the spectral interference of water and medium components.

**Table 2.** Prediction errors for partial least square regression (PLSR) and random forest regression (RFR) models.

| | | Baseline-Corrected | | | Baseline-Corrected (Post-Induction) | | |
|---|---|---|---|---|---|---|---|
| | | d0 | d1 | d2 | d0 | d1 | d2 |
| PLSR | Components | 6 | 2 | 5 | 6 | 2 | 2 |
| | | | | NRMSE | | | |
| | AP | 0.13 | 0.11 | 0.10 | 0.18 | 0.24 | 0.22 |
| | SpA | 0.28 | 0.35 | 0.23 | 0.22 | 0.27 | 0.26 |
| | DNA | 0.14 | 0.13 | 0.15 | 0.21 | 0.20 | 0.18 |
| RFR | Pruning | 0 | 0 | 0 | 0 | 0 | 0 |
| | | | | NRMSE | | | |
| | AP | 0.18 | 0.12 | 0.11 | 0.17 | 0.15 | 0.14 |
| | SpA | 0.28 | 0.21 | 0.22 | 0.32 | 0.25 | 0.21 |
| | DNA | 0.16 | 0.12 | 0.12 | 0.21 | 0.15 | 0.15 |

*3.4. Classification*

The issue of data scarcity was not given for the classification tasks since all spectra could be assigned to classes depending on whether thresholds in the reference data were exceeded. PLSDA, LDA, RFC, and ANN models were built for the supervised classification of normal, leaky, and lysing cells. Due to the high variance introduced by the spectral shift at the start of induction, only the baseline-corrected, post-induction spectra were used to train the models. The results for the initial grid-search assessing a wider set of parameters for the ANN model (hidden layer structure, number of epochs, weight penalization parameter) are summarized in Figure S2. For further analysis, ANN models with two hidden layers and 10 nodes each were trained for 75 epochs on the d0 and d1 dataset, and ANNs with two hidden layers and 200 nodes each were trained for 400 epochs on the d2 dataset. Figure 6 shows the results of the screening for the optimal regularization parameters for all models and each derivative applied during preprocessing. Among the linear models PLSDA and LDA, the latter showed significantly better performance with accuracies and $F_1$ scores for all classes above 0.8 when high shrinkage was applied. ANN models in combination with the first derivative performed slightly better than LDA, whereas the best overall scores were achieved with RFCs in combination with the first and second derivatives (d1, d2). They each achieved accuracies of above 0.93 and $F_1$ scores above 0.90 over a wide range of applied regularization parameters (Figure 6, Figure S3). Interestingly, the least regularized RFC and ANN models did not result in drastic performance reduction from overfitting, which could be well observed in LDA models. Nonetheless, regularization was applied to RFCs trained on d1 or d2 data for further analysis by setting the pruning parameter to 0.06.

To translate the classification performance of the RFC models to the domain of process time, the correct and false classifications of each spectrum from the respective test set were visualized in Figure 7. The timeframe in which wrong predictions were made varied between a few minutes (e.g., run LC3) and approximately an hour (runs LC4, LC5) and was similar between RFCs trained on the first or second derivative data. However, misclassifications occurred mostly during the transitions between normal, leaky, or lysis. As shown by PCA (Figure 4), this is likely due to smooth transitions in the spectra between the different cell phases, complicating classification tasks. Furthermore, the true labels were created by manual assignment based on absolute thresholds of the reference data (SpA and DNA concentration), which were collected in intervals between one and two hours. Particularly for run LC6, the threshold for lysis was estimated by interpolation due to a large gap in the reference data between five and eight hours after induction. Hence, any bias introduced by the manual label assignment is also reflected in the results.

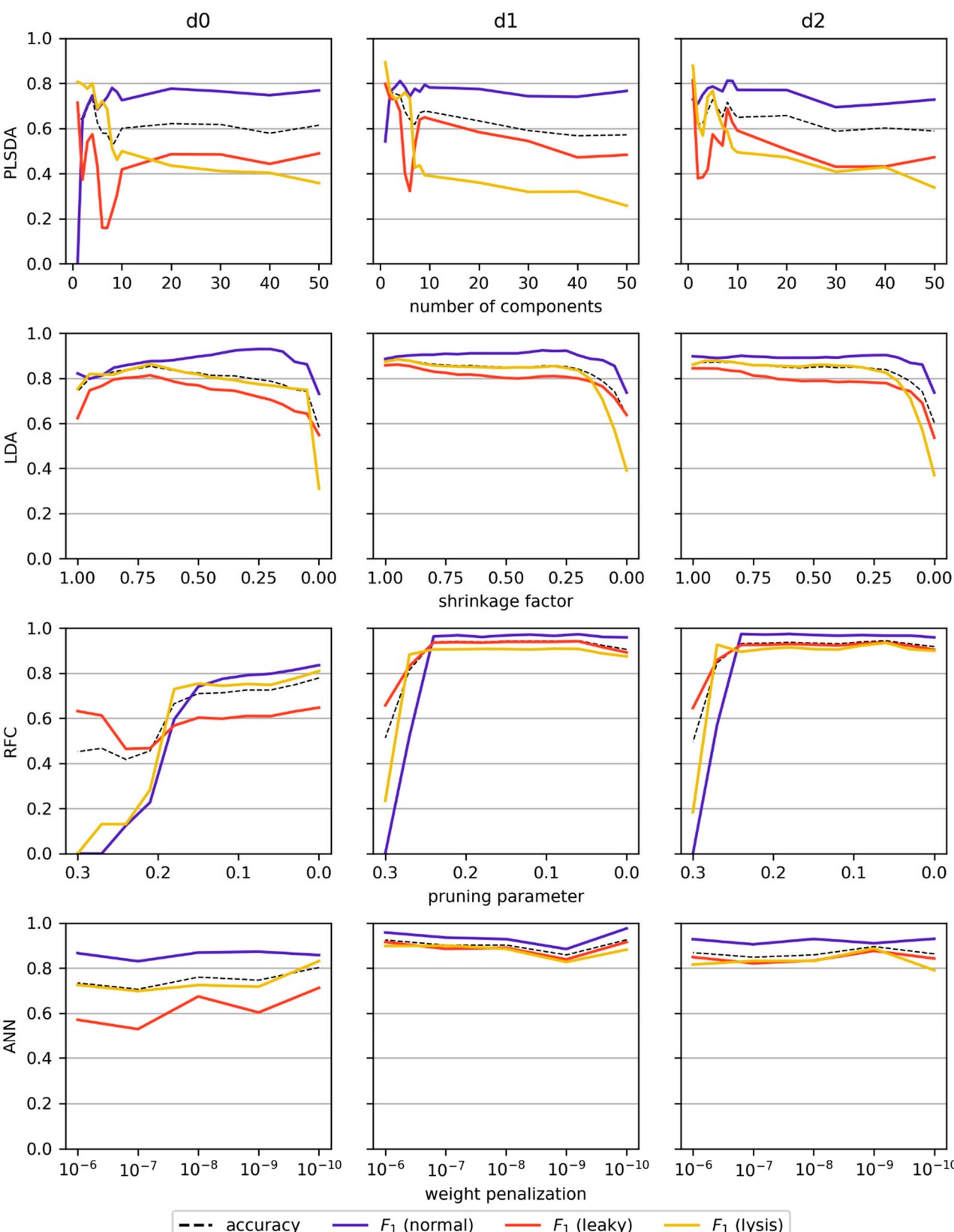

**Figure 6.** Classification performance of the different models trained and tested with the baseline-corrected post-induction spectra vs. different degrees of complexity. The x-axes are oriented such that the least complex model is on the left side and the most complex on the right side. The evaluation metrics were averaged over the six data splits (one for each run), weighted by the number of samples in each run (accuracy) or the frequency of classes in each run ($F_1$ score).

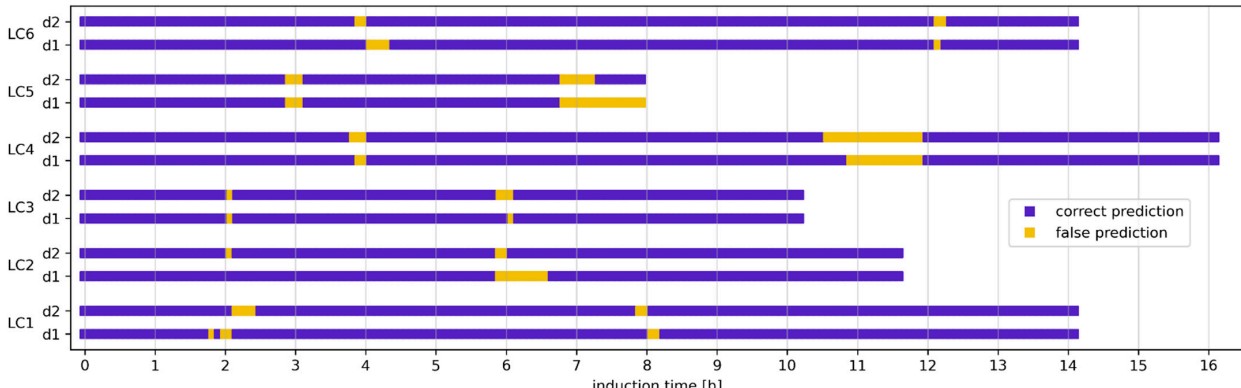

**Figure 7.** Correct and false classifications of the random forest classifiers (RFCs) in the time domain. Models were evaluated with the first and second derivatives (d1 and d2) of baseline-corrected post-induction spectra. The pruning parameter was set to 0.06.

The feature importance of the RFC models (Figure 8A) using the whole fingerprint region (850–1700 cm$^{-1}$) indicated that the amide I and II bands at 1655 and 1550 cm$^{-1}$ strongly contributed to classification. Moreover, bands at ~1000 and 1050 cm$^{-1}$ had a large impact as well. This points toward interference by spectral changes caused by the continuous dilution of medium components, especially L-arabinose, during the cultivation, although the distinct spectral feature of lysed cells at 1048 cm$^{-1}$ (Figure 3B) may have influenced the model as well. However, the prediction accuracy of the RFC was robust against feature selection (Figure 8B). Only retaining the amide I and II regions (sel-1) or removing the spectral ranges attributed to L-arabinose (sel-2), the amide I and II bands (sel-3), or both of these ranges (sel-4) did not result in a significant reduction of the classification accuracy. Even after extreme feature exclusion (sel-9; leaving only wavenumbers between 1360 and 1450 cm$^{-1}$), the model still classified approximately 80% of samples correctly. Feature selection was also applied to LDA and ANN models, where similar behavior was observed (Figure S4). This robustness against the removal of (apparently important) features shows that classification is strongly relying on correlations between the cell states (normal, leaky, lysis) and untargeted process dynamics reflected in the spectra.

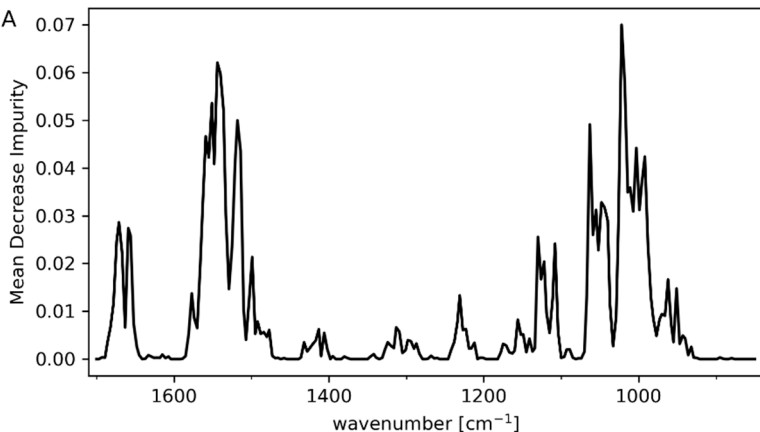

**Figure 8.** *Cont.*

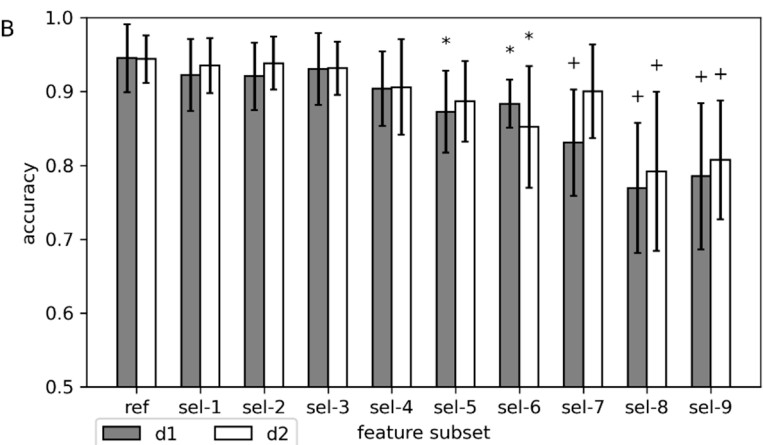

**Figure 8.** Feature importance and feature selection with RFC. (**A**) Feature importance calculated by MDI averaged over the six different data splits (one for each run) for RFC trained on first derivative data. (**B**) Classification accuracy after removal of features averaged over the six different data splits (one for each run), weighted by the number of samples in each run. Feature selection strategies are listed in Table 1. Significant difference in average accuracy to the reference (ref) is denoted by * ($p <$ 0.05) or + ($p <$ 0.01).

## 4. Discussion

In applications of in-line ATR sensors for bioprocessing, the small penetration depth of the evanescent field into the sample has been a hurdle in the quantification of biomass since cells barely reside in the absorbing volume close to the sensor surface without complex modification of the probe [15,16,44,45]. For monitoring leakiness and lysis, however, this property of the ATR sensor is advantageous, as it provides more selectivity toward changes in the culture supernatant, such as leaked periplasmic protein or cellular molecules released upon lysis. Our off-line experiments comparing the spectra of suspended and settling cells underlined this advantage. However, the sensitivity toward target compounds, most importantly protein, is severely hampered by the strong adsorption of water in the fingerprint region. This is a disadvantage of ATR-FTIR compared to Raman spectroscopy [17], although the spectral contribution of biomass is higher for conventional Raman spectroscopy, and probe modifications are required to ensure the separation of cells and medium. This was recently achieved with synthetic particles by ultrasound-enhanced in-line probes [46], but applicability to *E. coli* cells has not yet been shown. The strong absorption of medium components and their spectral overlap with lysed cells was another potential obstacle in the present work. Continuous dilution of the medium components by the fed-batch strategy introduces correlation with target analytes, possibly leading to predictions by chance [7]. Whereas the preprocessing steps employed in this study could reduce the between-run variance, the within-run variance was still dominated by these correlated process dynamics.

In addition to detection of the onset of leakiness and lysis, predicting the concentrations of target analytes, such as extracellular protein or DNA, by in-line spectroscopy can facilitate advanced, model-based process control [1]. PLSR is a well-established chemometric tool for MIR spectroscopy and has been frequently used for the quantification of small organic metabolites or inorganic medium components in bioprocesses [5–9]. In the present study, however, PLSR performed poorly for the prediction of extracellular SpA, AP, or DNA and was therefore not found suitable for monitoring. RFR, which was recently applied to the monitoring of animal cell cultures with Raman spectroscopy [47], did not achieve better results. The limited success of regression can be attributed to several factors. Firstly, the absorption of water greatly interferes with the amide bands. Hence, the sensitivity for protein is reduced, and any noise in the water spectrum produces an error in the prediction of protein concentration [48]. Moreover, AP and DNA were not directly predicted since their concentrations were likely too low to be detected. Thus, their concentration profile may

not correlate well with changes in the spectral fingerprint, and their prediction is largely based on correlation with other process dynamics. Lastly, the number of samples was too low for building a powerful statistical model given the complex sample matrix. Extending the calibration set with external synthetic samples [7] or implementing on-line reference analyses [6] are possible solutions to data scarcity—especially in academic settings, where usually less process data is generated compared to industry.

The problem of data scarcity could be circumvented by the classification of all available spectra based on thresholds in the reference data. The non-linear classifiers RFC and ANN showed overall better performance than the linear methods, which agrees with previous studies employing non-linear techniques on spectral data from bioprocesses [18–21]. Although the timeframe in which the RFC yielded wrong classifications was mostly under 20 min, in some cases, it was up to approximately one hour. Hence, the investigated method did not always provide faster results than available at-line techniques, such as high-performance liquid chromatography or colorimetric assays [49]. Furthermore, the caveats of using machine learning for the classification of in-line ATR-FTIR spectra were demonstrated. Despite removing the features deemed important for the task of predicting leakiness and lysis, and even after removing almost all available features from the data, classification models still displayed relatively high prediction accuracy. As discussed above, the correlation of the target variables, respectively class labels, with untargeted process dynamics can produce misleading results [7]. Thus, whereas the method performed well on the dataset used in this study, the transferability to a different process is questionable (e.g., a fermentation without leaky or lysing cells or a process with a different feeding strategy). To provide more confidence in the predictions of the classifiers and tackle the issue of strong correlation with process time rather than target features, the models would need to be presented with data from runs with a wider variety of process parameters (and thus more variation in process dynamics) or with synthetic, defined samples, e.g., by spiking a fermentation sample with cell lysate or product [7].

## 5. Conclusions

In summary, this study presents a novel approach to monitoring cell integrity during recombinant protein production with *E. coli* via in-line ATR-FTIR spectroscopy. Whereas regression models performed poorly due to low amounts of reference data and a possible lack of sensitivity, classification with LDA, RFC, and ANN resulted in high apparent prediction accuracy. RFCs achieved the best results and were able to predict changes in cell integrity in a timeframe between ~10 and up to ~70 min. However, classification results were strongly influenced by correlations with untargeted, time-dependent changes in the spectra. This demonstrates the need for rigorous model validation (including feature selection) to avoid misinterpretation of spectral data from bioprocesses, as well as the necessity for appropriate datasets with high variability in the targeted features. Further research may explore the potential of Raman spectroscopy, possibly in combination with ultrasound manipulation of cells, for monitoring *E. coli* cell integrity.

**Supplementary Materials:** The following are available online at https://www.mdpi.com/2227-9717/9/3/422/s1, Figure S1: Reference data from the induction phase of the six bioprocesses, Table S1: Number of samples and class frequency in each run, Figure S2: Hyperparameter grid search for ANN models, Figure S3: Magnification of Figure 6 in the main text, Figure S4: Classification accuracy of LDA and ANN after feature selection.

**Author Contributions:** Conceptualization, O.S. and J.K.; investigation, J.K. and J.L.-E.; data curation, J.K., J.L.-E., and V.R.; validation, J.K., J.L.-E., and V.R.; writing—original draft preparation, J.K. and J.L.-E.; writing—review and editing, O.S. and V.R.; visualization, J.K. and J.L.-E.; supervision, O.S. and V.R.; project administration, O.S.; funding acquisition, O.S. All authors have read and agreed to the published version of the manuscript.

**Funding:** This research was funded by the Austrian Research Promotion Agency (FFG), grant number 872643.

**Data Availability Statement:** The data presented in this study are available on request from the corresponding author.

**Acknowledgments:** The authors gratefully acknowledge Christoph Gasser for his support in the chemometric analysis. The authors thank the TU Wien Bibliothek for financial support through its Open Access Funding Program.

**Conflicts of Interest:** The authors declare no conflict of interest. The funders had no role in the design of the study; in the collection, analyses, or interpretation of data; in the writing of the manuscript; or in the decision to publish the results.

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
