# Peer review of "Monitoring E. coli Cell Integrity by ATR-FTIR Spectroscopy and Chemometrics: Opportunities and Caveats"

_processes, doi:10.3390/pr9030422_

Round 1

Reviewer 1 Report

The manuscript "Monitoring E. coli cell integrity by ATR-FTIR spectroscopy and chemometrics: opportunities and caveats" assessed the applicability of in-line ATR-FTIR spectroscopy for real-time monitoring of the OM and IM integrity of E. coli during recombinant protein production. It is well organized; however, the work is not adequate in my opinion. The authors should make a major revision before I can revise in future.

Since the authors mentioned “Many biomolecules display specific spectral features in the MIR range, 47 particularly in the so-called fingerprint region” and “distinct absorbance at the amide I and II bands (1655 and 1550 cm-1, respectively) as well as at 1077 cm-1 may be attributed to carbohydrates”, it cannot define the peaks in the results definitely caused by the recombinant protein of OM and IM. Although the authors test the BSA (bovine serum albumin) protein, the protein from cells or lysed cells are totally different from BSA protein. These peaks may be caused by others (such as DNA or sugars) after cell lysing. A dielectric spectroscopy should be performed as reference to verify the results.   

Author Response

Reviewer #1:

  • Since the authors mentioned “Many biomolecules display specific spectral features in the MIR range, 47 particularly in the so-called fingerprint region” and “distinct absorbance at the amide I and II bands (1655 and 1550 cm-1, respectively) as well as at 1077 cm-1 may be attributed to carbohydrates”, it cannot define the peaks in the results definitely caused by the recombinant protein of OM and IM. Although the authors test the BSA (bovine serum albumin) protein, the protein from cells or lysed cells are totally different from BSA protein. These peaks may be caused by others (such as DNA or sugars) after cell lysing.
    • Thank you for the valuable input. We acknowledge the fact that the specific fingerprint of coli proteins is different from BSA, as we mentioned in the text (lines 276 ff). However, the purpose of these off-line experiments with BSA was to explore important spectral features of dissolved protein for later comparison with inline spectra, as well as the sensitivity of the investigated method for protein. We rephrased the corresponding passage in the text (lines 270 f) for clarity. The amide I, II and III bands are known and conserved spectral features of proteins (further elaborated in lines 272 ff). Although their exact shape and intensity is dependent on the primary and secondary structure, we show that the amide bands are weak compared to medium components, particularly phosphate. We further explained this in lines 283 ff.
    • DNA and sugars mostly absorb at lower wavenumbers (ca 900-1200 cm-1; cf. Baker et al., 2016; Quintelas et al., 2018). In our off-line experiments, cell lysis produced a distinct peak at 1048 cm-1, which may be attributed to membrane components (such as lipopolysaccharides). We added an explanation to the text (lines 297 f.).

  • A dielectric spectroscopy should be performed as reference to verify the results.
    • Thank you for the comment. We measured DNA accumulation in the supernatant with the PicoGreen assay, which is a highly sensitive and specific method established for quantification of lysis (e.g. Sissolak et al. 2019, Newton et al. 2017) and therefore well-suited as a reference for this study.

References:

Baker, M. J., Hussain, S. R., Lovergne, L., Untereiner, V., et al., Developing and Understanding Biofluid Vibrational Spectroscopy: A Critical Review. Chem. Soc. Rev. 2016, 45, 1803-1818.

Newton, J. M., Vlahopoulou, J., Zhou, Y., Investigating and Modelling the Effects of Cell Lysis on the Rheological Properties of Fermentation Broths. Biochem. Eng. J. 2017, 121, 38-48.

Quintelas, C., Ferreira, E. C., Lopes, J. A., Sousa, C., An Overview of the Evolution of Infrared Spectroscopy Applied to Bacterial Typing. Biotechnol. J. 2018, 13, 1700449.

Sissolak, B., Zabik, C., Saric, N., Sommeregger, W., et al., Application of the Bradford Assay for Cell Lysis Quantification: Residual Protein Content in Cell Culture Supernatants. Biotechnol. J. 2019, 14, 1800714.

Reviewer 2 Report

In my opinion this is an interesting and novel paper to use the ATR-FTIR spectroscopic technology combing with some process analytical 24 technology to measure the E. coli cell integrity.
1. line 53: please define HCP in the first place it is proposed (line 37);
2. Line 137: is there a mistake about the unit of the data here?
3. For Figure 2, could you show how to do the measurement for the ATR probe immersed in a stirred cell suspension, is it possible to provide the structural scheme for this two kind of measurements by ATR, respectively? So it is more Intuitive. (this is just an optional suggestion)
4. For Figure 3A, could you please explain the intensity changes of the peaks from amide I-III with the different concentration of BSA solutions by one to two sentences?

Author Response

Reviewer #2:

  • line 53: please define HCP in the first place it is proposed (line 37)
    • Thank you for the comment, the definition was added.
  • Line 137: is there a mistake about the unit of the data here?
    • The yield coefficient YX/S is given in grams cell dry weight per gram glucose. Respective indices were added.
  • For Figure 2, could you show how to do the measurement for the ATR probe immersed in a stirred cell suspension, is it possible to provide the structural scheme for this two kind of measurements by ATR, respectively? So it is more Intuitive. (this is just an optional suggestion)
    • We added insets into the figure depicting the sensor setup.
  • For Figure 3A, could you please explain the intensity changes of the peaks from amide I-III with the different concentration of BSA solutions by one to two sentences?
    • We added a more detailed explanation in the text (lines 273 ff, 283 ff).

Round 2

Reviewer 1 Report

Agreed to be accepted.